# Quantum dynamics in transverse-field Ising models from classical networks

Markus Schmitt[1*] and Markus Heyl[2]

**1** Institute for Theoretical Physics, Georg-August-Universität Göttingen,
Friedrich-Hund-Platz 1 - 37077 Göttingen, Germany
**2** Max-Planck-Institute for the Physics of Complex Systems,
Nöthnitzer Str. 38 - 01187 Dresden, Germany

* markus.schmitt@theorie.physik.uni-goettingen.de

## Abstract

The efficient representation of quantum many-body states with classical resources is a key challenge in quantum many-body theory. In this work we analytically construct classical networks for the description of the quantum dynamics in transverse-field Ising models that can be solved efficiently using Monte Carlo techniques. Our perturbative construction encodes time-evolved quantum states of spin-1/2 systems in a network of classical spins with local couplings and can be directly generalized to other spin systems and higher spins. Using this construction we compute the transient dynamics in one, two, and three dimensions including local observables, entanglement production, and Loschmidt amplitudes using Monte Carlo algorithms and demonstrate the accuracy of this approach by comparisons to exact results. We include a mapping to equivalent artificial neural networks, which were recently introduced to provide a universal structure for classical network wave functions.

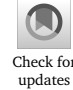

# 1   Introduction

A key challenge in quantum many-body theory is the efficient representation of quantum many-body states using classical compute resources. The full information contained in such a many-body state in principle requires resources that grow exponentially with the number of degrees of freedom. Therefore, reliable schemes for the compression and efficient encoding of the essential information are vital for the numerical treatment of correlated systems with many degrees of freedom. This is of particular relevance for dynamics far from equilibrium, where large parts of the spectrum of the Hamiltonian play an important role.

For low-dimensional systems matrix product states [1,2] and more general tensor network states [3] constitute a powerful ansatz for the compressed representation of physically relevant many-body wave functions. These allow for the efficient computation of ground states and real time evolution. In high dimensions properties of quantum many-body systems in and out of equilibrium can be obtained by dynamical mean field theory [4–7], which yields exact results in infinite dimensions. This leaves a gap at intermediate dimensions, where exciting physics far from equilibrium has recently been observed experimentally [8–13].

An alternative approach, which received increased attention lately, is the representation of the wave function based on networks of classical degrees of freedom. Given the basis vectors $|\vec{s}\rangle = |s_1\rangle \otimes |s_2\rangle \otimes \ldots \otimes |s_N\rangle$ of a many-body Hilbert space, where the $s_l$ label the local basis, the coefficients of the wave function $|\psi\rangle$ are expressed as

$$\psi(\vec{s}) = \langle \vec{s}|\psi\rangle = e^{\mathscr{H}(\vec{s})}, \tag{1}$$

where $\mathscr{H}(\vec{s})$ is an effective Hamilton function defining the classical network. Wave functions of this form were used in combination with Monte Carlo algorithms for variational ground state searches [14–16] and time evolution [17–23]. Recently, it was suggested that the wave function (1) can generally be encoded in an artificial neural network (ANN) trained to resemble the desired state [23]. This idea was seized in a series of subsequent works exploring the capabilities of this and related representations [24–31]. Importantly, there are no principled restrictions on dimensionality.

In this work we present a scheme to perturbatively derive analytical expressions for perturbative classical networks (pCNs) as representation of time-evolved wave functions for transverse-field Ising models (TFIMs) which can be extended directly also to other models. The resulting networks consist of the same number of classical spins as the corresponding quantum system and exhibit only local couplings making the encoding particularly efficient. We compute the transient dynamics of the TFIM in one, two, and three dimensions ($d = 1, 2, 3$) including local observables, correlation functions, entanglement production, and Loschmidt amplitudes. By comparing to exact solutions we demonstrate the accuracy of our results going well beyond standard perturbative approaches. This work provides a way to derive classical network structures within a constructive prescription, where other approaches rely on heuristics. As a specific application, we derive the structure and the time-dependent weights of equivalent ANNs in the sense of Ref. [23].

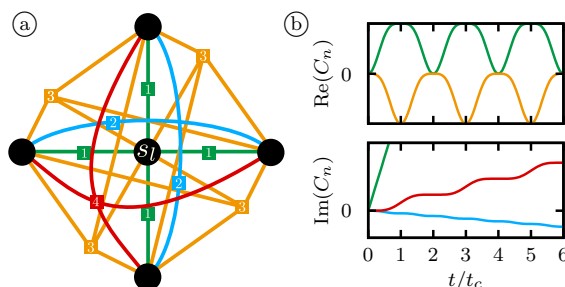

Figure 1: (a) Structure of the perturbative classical network for the TFIM in $d = 2$ and (b) dynamics of the couplings (color coded as in (a)). The black dots in the network structure represent a classical spin $s_l$ and its four neighbors in a translationally invariant square lattice. Each square with number $n$ stands for a coupling of the connected classical spins with coupling constant $C_n(t)$. The green and blue lines, respectively, correspond nearest-neighbor and next-nearest-neighbor coupling of two spins, while the orange and red lines indicate coupling terms involving four spins each. The resulting time-dependent classical Hamiltonian function $\mathcal{H}(\vec{s}, t)$ encodes quantum dynamics via Eq. (1).

## 2 Results

In the following we compute dynamics of TFIMs of $N$ spins with Hamiltonian

$$H = -\frac{J}{4} \sum_{\langle i,j \rangle} \sigma_i^z \sigma_j^z - \frac{h}{2} \sum_{i=1}^{N} \sigma_i^x \, , \tag{2}$$

where $\sigma_i^{x/z}$ denote Pauli operators acting on site $i$ and the first sum runs over neighboring lattice sites $i$ and $j$. As the computational basis we choose the spin basis states $|\vec{s}\rangle = |s_1 \ldots s_N\rangle$ with $s_i = \uparrow, \downarrow$. The dynamics of Ising models is accessible experimentally with quantum simulators, which was demonstrated recently in various setups [32–34]. In $d = 1$ the dynamics of the TFIM can be computed analytically by means of a Jordan-Wigner transform [35–44].

In this work we are interested in the dynamics that comprise a dynamical quantum phase transition (DQPT) [45, 46]. The signature of a DQPT is a non-analyticity in the many-body dynamics analogous to equilibrium phase transitions where thermodynamic quantities behave non-analytically as function of a control parameter. DQPTs were recently observed in experiment [11, 34] and there is a series of results on TFIMs in this context [47–57].

Typically, DQPTs occur when the model is quenched across an underlying equilibrium quantum phase transition. A particularly insightful limit with this respect is a quench from $h_0 = \infty$ to $h/J \ll 1$, where, e.g., universal behavior was proven in $d = 1$ [51]. When quenching from $h_0 = \infty$ to $h = 0$ the TFIM in $d = 1, 2$ exhibits DQPTs at odd multiples of $t_c = \pi/J$, which we choose as the unit of time throughout the paper. The ground state at $h_0 = \infty$ is a particularly simple initial state, since $\langle \vec{s} | \psi_0 \rangle = 2^{-N/2}$. One could, however, go away from that limit perturbatively, e.g., by constructing a Schrieffer-Wolff transformation for an initial state with weak spin couplings.

Quench dynamics of the two-dimensional TFIM have already been studied in Refs. [20, 21], but there quenches within the same phase have been considered in contrast to the extreme quench across the phase boundary, which we will address in the following.

### 2.1 Classical network via cumulant expansion

Consider a Hamiltonian of the form $H = H_0 + \lambda V$, where $H_0$ is diagonal in the spin basis, $H_0 | \vec{s} \rangle = E_{\vec{s}} | \vec{s} \rangle$, $V$ an off-diagonal operator, and $\lambda \ll 1$. In the interaction

picture the time evolution operator can be expressed as $e^{-iHt} = e^{-iH_0 t} W_\lambda(t)$, where $W_\lambda(t) = \mathcal{T}_t \exp\left[-i\lambda \int_0^t dt' V(t')\right]$. In this setting time-evolved coefficients of the wave function (1) can be obtained perturbatively by a cumulant expansion [58]. Denoting the initial state with $|\psi_0\rangle = \sum_{\vec{s}} \psi_0(\vec{s}) |\vec{s}\rangle$ the cumulant expansion to lowest order yields the time-evolved state $|\psi(t)\rangle = \sum_{\vec{s}} \psi(\vec{s}, t)|\vec{s}\rangle$ with

$$\frac{\psi(\vec{s}, t)}{\psi_0(\vec{s})} = e^{-iE_{\vec{s}} t} \exp\left[-i\lambda \int_0^t dt' \frac{\langle \vec{s}|V(t')|\psi_0\rangle}{\langle \vec{s}|\psi_0\rangle} + \mathcal{O}(\lambda^2)\right]. \tag{3}$$

By identifying $\mathcal{H}(\vec{s}, t) = -iE_{\vec{s}} t - i\lambda \int_0^t dt' \frac{\langle \vec{s}|V(t')|\psi_0\rangle}{\langle \vec{s}|\psi_0\rangle}$ the expression above takes the desired form given in Eq. (1). Importantly, also the effective Hamilton function becomes local, whenever $H_0$ and $V$ are local. It will be demonstrated below that the construction via cumulant expansion yields much more accurate results than conventional perturbation theory. The approximation can be systematically improved by taking into account higher order terms. To which extent it is possible to also capture long-time dynamics using such a construction, remains an open question and, since beyond the scope of the present work, will be left for future research.

For our purposes, we identify $H_0 = -\frac{J}{4} \sum_{\langle i,j \rangle} \sigma_i^z \sigma_j^z$ and $\lambda V \hat{=} -\frac{h}{2} \sum_i \sigma_i^x$. Note that, e.g., a strongly anisotropic XXZ model could be treated analogously. The time-dependent $V(t)$ is obtained by solving the Heisenberg equation of motion. The general form of the Hamilton function from the first-order cumulant expansion obtained under these assumptions is

$$\mathcal{H}^{(1)}(\vec{s}, t) = \sum_{n=0}^{z} C_n(t) \sum_{l=1}^{N} \sum_{(a_1, \ldots, a_n) \in \mathcal{V}_n^l} s_l^n \prod_{r=1}^{n} s_{a_r}, \tag{4}$$

where $\mathcal{V}_n^l$ denotes the set of possible combinations of $n$ neighboring sites of lattice site $l$, $z$ is the coordination number of the lattice, and $C_n(t)$ are time-dependent complex couplings. Classical Hamilton functions $\mathcal{H}^{(1)}(\vec{s}, t)$ for cubic lattices in $d = 1, 2, 3$ including explicit expressions for the couplings $C_n(t)$ are given in Appendix A. Fig. 1 displays the structure of the pCN in 2D and the time evolution of the couplings $C_n(t)$. For $d = 2, 3$ $\mathcal{H}^{(1)}(\vec{s}, t)$ already contains couplings with products of four or six spin variables, respectively. Thereby, the derived structure of the pCN markedly differs from heuristically motivated Jastrow-type wave functions, which constitute a common variational ansatz [17, 20]. From our perturbative construction we find that it is already at lowest order important to take into account plaquette interactions of more than two spins in order to obtain accurate results.

The data we present in the following were obtained with $h/J = 0.05$. Results for larger $h/J$ are presented in Appendix A. There we find that comparable accuracy is obtained for times $ht \ll 1$. As we will show in the following the accuracy can be enhanced by including higher order contributions from the cumulant expansion. However, the resulting coupling parameters $C_n(t)$ comprise secular terms, which grow with increasing time. We anticipate that these secular terms restrict the time-window in which the couplings obtained from the cumulant expansions yield precise results to $t < h^{-1}$. Nevertheless, we expect that an effective resummation of secular contributions can be achieved by combining the perturbatively derived network structures with a time-dependent variational principle [17, 59–61].

## 2.2 Observables

Plugging Eq. (1) into the time-dependent expectation value of an observable $\hat{O}$ with matrix elements $\langle \vec{s}|\hat{O}|\vec{s}'\rangle = O_{\vec{s}} \delta_{\vec{s},\vec{s}'}$ results in

$$\langle \psi_0 | e^{iHt} \hat{O} e^{-iHt} | \psi_0 \rangle = \sum_{\{\vec{s}\}} e^{\mathcal{H}(\vec{s}, t)} \tilde{O}_{\vec{s}}. \tag{5}$$

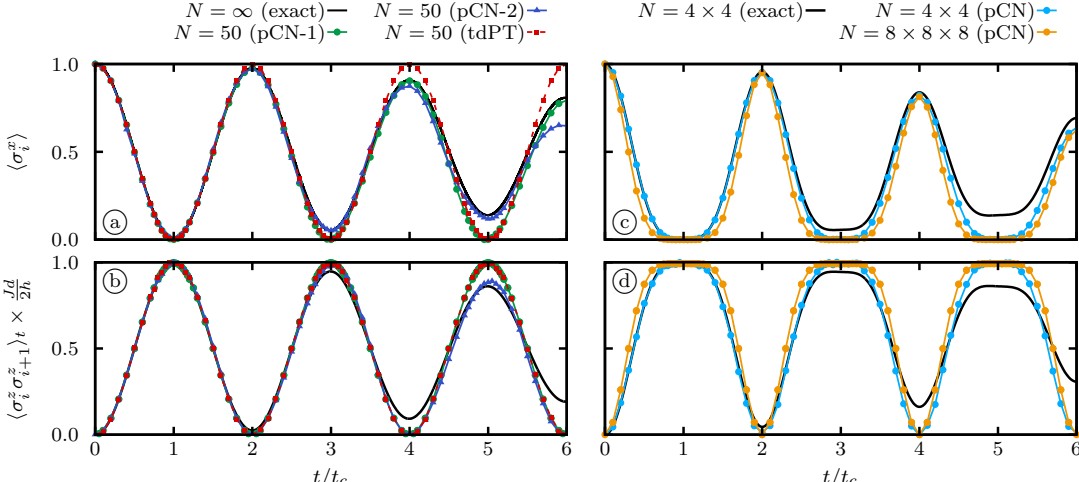

Figure 2: Time evolution of transverse magnetization (top panels) and nearest-neighbor correlation function (bottom panels) in the TFIM. (a, b) Results for $d = 1$ obtained from the pCN with first order (pCN-1) and second order (pCN-2) expansion in comparison with the exact dynamics and time-dependent perturbation theory (tdPT). (c, d) Dynamics in $d = 2$ (blue), and $d = 3$ (orange) obtained from the first order pCN compared to exact results in $d = 2$. Data obtained with $h/J = 0.05$; $t_c = \pi/J$.

with

$$\tilde{O}_{\vec{s}} = \sum_{\{\vec{s}'\}} \mathrm{Re}\left[ O_{\vec{s}\vec{s}'} e^{\mathscr{H}(\vec{s}',t) - \mathscr{H}(\vec{s},t)} \right] \tag{6}$$

and $\tilde{\mathscr{H}}(\vec{s},t) = 2\,\mathrm{Re}[\mathscr{H}(\vec{s},t)]$. In this form the quantum expectation value resembles a thermal expectation value in the pCN defined by $\mathscr{H}(\vec{s},t)$. For an observable $\hat{O}$ that is diagonal in the spin basis, $\langle \vec{s}|\hat{O}|\vec{s}'\rangle = O_{\vec{s}}\delta_{\vec{s},\vec{s}'}$, the expression above simplifies to

$$\langle \psi_0 | e^{\mathrm{i}Ht} \hat{O} e^{-\mathrm{i}Ht} | \psi_0 \rangle = \sum_{\{\vec{s}\}} e^{\tilde{\mathscr{H}}(\vec{s},t)} O_{\vec{s}} \,. \tag{7}$$

These expressions can be evaluated efficiently by the Metropolis algorithm [62]. Although we find empirically that the off-diagonal observables under consideration can still be sampled efficiently by Monte Carlo, it is not clear whether a sign problem can appear in other cases. Fig. 2 shows results for different local observables obtained in this way. In these and the following figures the Monte Carlo error is less than the resolution of the plot.

In Fig. 2(a,b) we compare the results from the classical network construction to exact results obtained by fermionization for the infinite system in $d = 1$ [35–44]. Focusing for the moment on the transverse magnetization $\sigma_i^x$ in Fig. 2(a) we find that on short times the pCN gives an accurate description of the dynamics. Upon improving our pCN construction by including the second-order contributions in the cumulant expansion, the time scale up to which the pCN captures quantitatively the real-time evolution of $\sigma_i^x$ increases suggesting that the expansion can be systematically improved by including higher order terms. For a further benchmarking of our results we also compare the pCN results to conventional first-order time-dependent perturbation theory. Clearly, the first-order pCN provides a much more accurate approximation to the exact dynamics, which originates in an effective resummation of an infinite subseries of terms appearing in conventional time-dependent perturbation theory. In Fig. 2(b) we consider the nearest-neighbor longitudinal correlation function $\sigma_i^z \sigma_{i+1}^z$ which is an observable diagonal in the spin basis. Compared to the offdiagonal observable studied in

**SciPost** SciPost Phys. 4, 013 (2018)

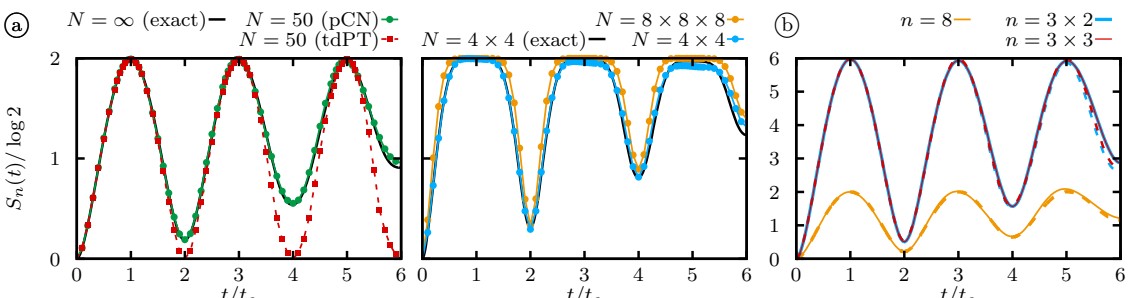

Figure 3: (a) Time evolution of the entanglement entropy for subsystems of $n = 2$ spins obtained from the classical network by MC in comparison with exact results; $h/J = 0.05$. (b) Time evolution of the entanglement entropy for different subsystem shapes with $n$ spins obtained from full wave functions $|\psi(t)\rangle$ determined from the pCN in comparison with exact results (dashed lines). In $d = 1$ the system size is $N = 20$, in $d = 2$ it is $N = 6 \times 3$; $h/J = 0.05$.

Fig. 2a we find much stronger deviations from the exact result which also cannot be improved upon including higher orders in the cumulant expansion. However, for correlation functions at longer distances the corrections to the first-order cumulant expansion become important; see Appendix A. The observation that the diagonal observables don't improve with the order of the pCN expansion we attribute to secular terms from resonant processes which are not appropriately captured by perturbative approaches such as the pCN. One possible strategy to incorporate such resonant processes is to impose a time-dependent variational principle [17, 59–61] on our networks in order to obtain suitably optimized coupling coefficients. Having demonstrated under which circumstances the pCN can be improved by including higher order contributions, for the remainder of the article we focus on the capabilities of the first-order pCN leaving further optimization strategies of the network open for the future.

In Fig. 2(c,d) we show our results for the same observables but now in $d = 2$ and $d = 3$. Compared to $d = 1$ we find much broader maxima and minima, respectively, close to the times where DQPTs occur at odd multiples of $t_c = \pi/J$. In the limit $h/J \to 0$ the shape is given by the power law $|t - t_c|^z$ with $z = 2d$. This behavior was already observed for one and two dimensional systems in Ref. [51]. For the $d = 2$ case we have included also exact diagonalization data for a $4 \times 4$ lattice. Overall, we observe a similar accuracy in the dynamics of these observables as compared to the $d = 1$ results.

## 2.3 Entanglement

Having discussed the capabilities of the pCN to encode the necessary information for the dynamics of local observables and correlations, we would like to show now that it can also reproduce entanglement dynamics and thus the propagation of quantum information.

By sampling all correlation functions it is in principle possible to construct the reduced density matrix of a subsystem $A$, $\rho_A(t) = \mathrm{tr}_B(|\psi(t)\rangle\langle\psi(t)|)$, where $\mathrm{tr}_B$ denotes the trace over the complement of $A$, and the entanglement entropy of subsystem $A$ given by $S(t) = -\mathrm{tr}(\rho_A(t)\ln\rho_A(t))$. For subsystems with two spins at sites $i$ and $j$ we have $\rho_A = \frac{1}{4}\sum_{\alpha,\alpha'\in\{0,x,y,z\}}\langle\sigma_i^\alpha\sigma_j^{\alpha'}\rangle\,\sigma^\alpha\otimes\sigma^{\alpha'}$, where $\sigma_i^0$ denotes the identity. This approach is in principle applicable to arbitrary subsystem sizes; however, it quickly becomes unfeasible, because the number of correlation functions that has to be sampled grows exponentially with subsystem size. In order to obtain insights into the entanglement properties of larger subsystems it might be possible to use the algorithm introduced in Ref. [63] for quantum Monte Carlo, which, however, is beyond the scope of this work. For small system sizes entanglement entropy

for any block size can be extracted directly from the full wave function as described below.

Figure 3(a) shows the entanglement entropy $S_2(t)$ of two neighboring spins. We find very good agreement of the Monte Carlo data based on the first-order cumulant expansion with the exact results. In particular, for the entanglement entropy the classical network captures both the decay of the maxima close to the critical times $(2n+1)t_c$ and the increase of the minima. As for the observables the shape in the vicinity of the maxima depends on $d$ and is for $h/J \to 0$ given by the same power laws. Note, that the pCN correctly captures the maximal possible entanglement $S_2^{\max} = 2\ln 2$. By contrast, the result from tdPT completely misses the decay of the oscillations.

In order to assess the capability of the pCN to capture the entanglement dynamics of larger subsystems we compute the whole wave function $|\psi(t)\rangle = \sum_{\vec{s}} \psi(\vec{s})|\vec{s}\rangle$ with the coefficients $\psi(\vec{s})$ as given in Eq. (3) for feasible system sizes. The entanglement entropy of arbitrary bipartitions is then obtained by a Schmidt decomposition. Fig. 3(b) shows entanglement entropies obtained in this way for subsystems of different sizes $n$ in $d = 1, 2$. The results imply that at these short times only spins at the surface of the subsystem become entangled with the rest of the system. The maxima for a subsystem of $n = 8$ spins in a ring of $N = 20$ spins in $d = 1$ lie close to $2\ln 2$, the theoretical maximum for the entanglement entropy of the two spins, which sit at the surface. This interpretation is supported by the results for a torus of $N = 6 \times 3$ spins with subsystems of size $n = 3 \times 2$ and $n = 3 \times 3$. In that case the entanglement entropy reaches maxima of $6\ln 2$, corresponding to 6 spins at the boundary. In both cases the results agree well with the exact results for times $t < 4t_c$. This again reflects the fact that the pCN from first-order cumulant expansion yields a good approximation of the dynamics of neighboring spins.

## 2.4 Loschmidt amplitude

Next, we aim to show that not only local but also global properties are well-captured by the classical networks. For that purpose we study the Loschmidt amplitude $\langle \psi_0 | \psi(t) \rangle$, which constitutes the central quantity for the anticipated DQPTs and which has been measured recently experimentally in different contexts [34, 64]. For a quench from $h_0 = \infty$ to $h = 0$ the Loschmidt amplitude

$$Z(t) = \frac{1}{2^N} \sum_{\vec{s} \in \{\pm 1\}^N} e^{i\frac{J}{4}t \sum_{\langle i,j \rangle} s_i s_j} \tag{8}$$

resembles the partition sum of a classical network with imaginary temperature $\beta = -\mathrm{i}t$ [51]. This expression is not suited for MC sampling because all weights lie on the unit circle in the complex plane rendering importance sampling impractical and indicating a severe sign problem. These issues can be diminished by constructing an equivalent network with real weights. After integrating out every second spin on the sublattice $\Lambda$, equivalent to one decimation step [65], the partition sum takes the form

$$Z(t) = \frac{1}{2^N} \sum_{\vec{s} \in \{\pm 1\}^{N/2}} \prod_{i \in \Lambda} 2\cos\left(\frac{J}{4}t \sum_{\langle i,j \rangle} s_j\right). \tag{9}$$

Choosing a suited ansatz the partition sum can be rewritten as $Z(t) = \sum_{\vec{s}} e^{\mathscr{H}(\vec{s},t)}$ with real Boltzmann weights given by an effective Hamilton function $\mathscr{H}(\vec{s}, t)$ that defines the classical network [51, 65, 66]. Generally, the effective Hamilton function takes the form

$$\mathscr{H}(\vec{s}, t) = \sum_{n=0}^{z/2} C_n(t) \sum_{l \in \Lambda} \sum_{(a_1, \ldots, a_{2n}) \in \mathscr{V}_{2n}^l} \prod_{r=1}^{2n} s_{a_r}. \tag{10}$$

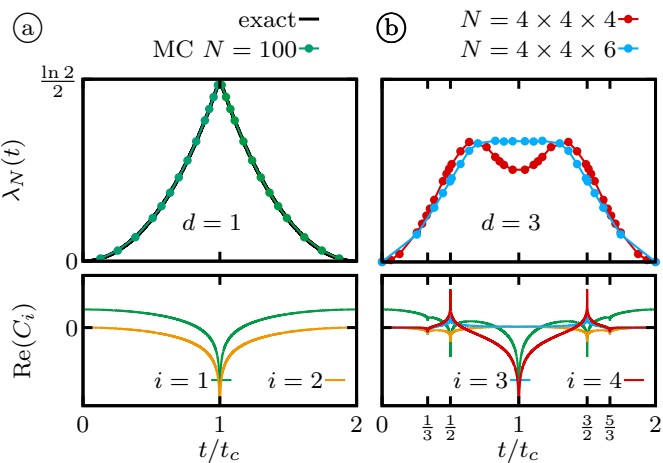

Figure 4: Time evolution of the rate function of the Loschmidt amplitude $\lambda_N(t)$ (top panels) and corresponding couplings in the classical network (bottom panels); (a) d=1, (b) d=3.

The explicit expressions for $d = 1, 2, 3$ are given in Appendix B.

It is evident from Eq. (9) that, although real, the Boltzmann weights of the classical network are not necessarily positive. Note that the absence of imaginary parts in the weights is due to the particular form of the Hamiltonian. For example, a nonvanishing transverse field would introduce imaginary parts and thereby complicate efficient Monte Carlo sampling. The bottom panels in Fig. 4 show the real parts of the coupling constants of the effective Hamiltonians for $d = 1, 3$. The couplings in $d = 3$ acquire non-vanishing imaginary parts for $t_c/3 \leq t \leq 5t_c/3$ leading to negative weights for some configurations. The partition sum is then split into a positive and a negative part $Z(t) = Z_+(t) + Z_-(t)$ with $Z_+ > 0$ and $Z_- < 0$. It was pointed out in Ref. [67] that the partition sum of such a factorized configuration space can be sampled despite the occurence of negative weights if the partial sums $Z_\pm$ can be sampled separately. In practice we perform separate Monte Carlo sampling on the respective configuration subspaces by prohibiting updates that change the sign of the weight. We combine this approach with parallel tempering [68] and multi-histogram reweighting [69] in order to render the sampling efficient and, moreover, to achieve the correct normalization. The proper normalization is crucial because $Z(t)$ is a quantum mechanical overlap. A more detailed description of the Monte Carlo scheme is given in Appendix B.

As the Loschmidt amplitude is exponentially suppressed with increasing system size we study the rate function [45] $\lambda_N(t) = -\frac{1}{N} \ln |Z(t)|$, which is well defined in the thermodynamic limit $N \to \infty$. The top panel in Fig. 4(a) displays $\lambda_N(t)$ obtained by a Monte Carlo sampling for a ring of $N = 100$ spins together with the exact result [70], confirming the precision of the pCN approach and demonstrating the principled possibility to detect DQPTs. For the rate function in $d = 3$ shown in Fig. 4(b) we obtained converged results in the whole interval for $N = 4 \times 4 \times 4$ and $N = 4 \times 4 \times 6$ physical spins. Note that there are no indications of non-analytic behavior in the Monte Carlo results at $t = t_c/3, t_c/2$ despite the divergences of the couplings at those points. While we can reach fairly large systems in $d = 3$, these are still not large enough to see convergence and non-analytic behavior at $t = t_c$ as opposed to the case of $d = 1$. It can be shown, see Appendix B, that for any dimension $\lambda_\infty(t_c) = \ln(2)/2$ demonstrating that our data in $d = 3$ is still far from the thermodynamic limit.

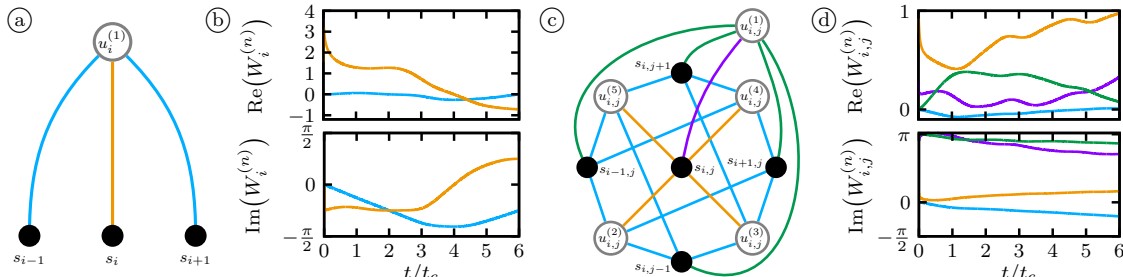

Figure 5: Structure of the ANN for the TFIM in $d = 1, 2$ (a, c) and time evolution of the weights obtained by first-order cumulant expansion for $h/J = 0.05$ (b, d). In the networks black dots stand for physical spins and gray circles indicate hidden spins. The couplings in (b, d) are color coded with the corresponding lines in (a, c).

## 2.5 Construction of equivalent ANNs

Finally, we present an exact mapping of the pCN obtained by a cumulant expansion to an equivalent ANN as introduced in Ref. [23]. This outlines the general potential of the pCN to guide the choice of network structures, for which otherwise no generic principle exists. Since the mapping is exact, observables sampled from the resulting network will be identical with the ones obtained from the pCN.

Generally, for Ising systems with translational invariance and local interactions, the cumulant expansion will yield a Hamilton function of the form

$$\mathscr{H}(\vec{s}, t) = \sum_{l=1}^{N} \mathscr{P}_l(\vec{s}, t), \tag{11}$$

where the functions $\mathscr{P}_l(\vec{s}, t)$ only involve a couple of spins in the neighborhood of spin $l$. We call the spins involved in $\mathscr{P}_l(\vec{s}, t)$ a *patch*. The $\mathscr{P}_l(\vec{s}, t)$ are invariant under $\mathbb{Z}_2$ and a number of permutations of the spins in a patch due to the lattice symmetries. In terms of the $\mathscr{P}_l(\vec{s}, t)$ the coefficients of the wave function are given by

$$\psi(\vec{s}, t) = e^{\mathscr{H}(\vec{s}, t)} = \prod_{l=1}^{N} e^{\mathscr{P}_l(\vec{s}, t)}. \tag{12}$$

To find the corresponding ANN we choose a general $\mathbb{Z}_2$ symmetric ansatz [23]

$$\psi_{ANN}(\vec{s}, t) = \left(\frac{\Omega}{2^\alpha}\right)^N \sum_{\vec{u}_l^{(1)} \dots \vec{u}_l^{(N_u)}} e^{\sum_{l,m} \sum_n W_{lm}^{(n)}(t) s_m u_l^{(n)}} \tag{13}$$

incorporating lattice symmetries in the connectivity of physical spins $s_l$ and hidden spins $u_l^{(n)}$ defined by the weights $W_{lm}^{(n)}$. $\alpha$ denotes the number of hidden spins per physical spin and $\Omega$ constitues an overall normalization. Upon integrating out the hidden spins we obtain

$$\psi(\vec{s}, t) = \prod_{l=1}^{N} \prod_{n=1}^{\alpha} \cosh\left(\sum_m W_{lm}^{(n)} s_m\right). \tag{14}$$

In order to determine the ANN weights we factor-wise equate the r.h.s. of Eq. (12) and Eq. (14),

$$\prod_n \cosh\left(\sum_m W_{lm}^{(n)} s_m\right) = e^{\mathscr{P}_l(\vec{s}, t)}, \tag{15}$$

and plug in each of the distinct spin configurations of a patch. This yields a set of equations for the unknown weigths $W_{lm}^{(n)}$, which can be solved numerically. In Appendix C procedure is outlined in detail for $d = 1$ and $d = 2$.

Fig. 5 shows the structure of the ANNs and the time-dependence of the weights obtained in this way for $d = 1$ and $d = 2$. In $d = 1$ the ANN structure (Fig. 5(a)) comprises the minimal number of hidden spins that is possible subject to the lattice symmetries. Although unproven the same is expected to hold for the structure for $d = 2$ in Fig. 5(c). Note the complex dynamics and the rapid initial change exhibited by some of the couplings. In comparison to a general all-to-all ansatz this construction provides a way to drastically reduce the number of ANN couplings in a controlled way, thereby restricting the variational subspace and lessening the computational cost for the optimization in variational algorithms.

## 3 Conclusions

In this work we introduced a perturbative approach based on a cumulant expansion that constitutes a constructive prescription to derive classical networks encoding the time-evolved wave function. The resulting pCNs are equivalent to corresponding ANNs, which were recently proposed as efficient representation of many-body states in Ref. [23]. For the quench parameters under consideration the pCNs give a good approximation of the initial dynamics and thereby provide a controlled benchmark for new algorithms targeting the dynamics in higher dimensions. In future work it is worth to explore whether the structure of the networks derived in this way constitutes a good ansatz for numerical time evolution based on a variational principle also in the absence of a small parameter [17, 59–61]. We expect that a variational time evolution based on the derived network structures could effectively perform the resummation of higher orders that would be necessary to overcome the problem of secular terms in the perturbative results. Moreover, the presented approach can be straightforwardly generalized to other systems and higher spin degrees of freedom. This might be particularly interesting in many-body-localized systems [9, 71–74], where the so-called local integrals of motion provide a natural basis for constructing a classical network.

## Acknowledgements

The authors acknowledge helpful discussions with S. Kehrein and M. Behr. For the numerical computations the Armadillo library [75] was used. The iTEBD algorithm was implemented using the iTensor library [76].

**Funding information**   M.S. is supported by the Studienstiftung des Deutschen Volkes. M.H. acknowledges support by the Deutsche Forschungsgemeinschaft via the Gottfried Wilhelm Leibniz Prize program.

# A  Perturbative classical networks

## A.1  Explicit expressions for the perturbative classical networks

For the cumulant expansion the time-evolved operator $V(t) = e^{iH_0 t} V e^{-iH_0 t}$ is required. This can be obtained by solving the corresponding Heisenberg equation of motion $-i\frac{d}{dt}V(t) = [H_0, V(t)]$.

In 1D the Heisenberg EOM for $\sigma_l^x(t)$ yields

$$\sigma_l^x(t) = \cos^2(Jt/2)\sigma_l^x - \sigma_{l-1}^z\sigma_{l+1}^z \sin^2(Jt/2)\sigma_l^x - i\frac{1}{2}\sin(Jt)\left(\sigma_{l-1}^z + \sigma_{l+1}^z\right)\sigma_l^z\sigma_l^x \ . \quad (16)$$

The cumulant expansion to first-order results in classical Hamilton functions of the general form

$$\mathscr{H}^{(1)}(\vec{s}, t) = -iE_{\vec{s}}t - i\lambda\sum_l \int_0^t dt' \frac{\langle \vec{s}|V(t')|\psi_0\rangle}{\langle \vec{s}|\psi_0\rangle} = \sum_{n=0}^z C_n(t)\sum_{l=1}^N \sum_{(a_1,\ldots,a_n)\in\mathscr{V}_n^l} s_l^n\prod_{r=1}^n s_{a_r} \ , \quad (17)$$

where $\mathscr{V}_n^l$ denotes the set of possible combinations of $n$ neighboring sites of lattice site $l$, $z$ is the coordination number of the lattice, and $C_n(t)$ are time-dependent complex couplings.

In $d = 1$ the explicit form is

$$\mathscr{H}_{1D}^{(1)} = NC_0(t) + C_1(t)\sum_l \left(s_{l-1}^z s_l^z + s_l^z s_{l+1}^z\right) + C_2(t)\sum_l s_{l-1}^z s_{l+1}^z, \quad (18)$$

with

$$\begin{aligned}
C_0(t) &= i\frac{h}{4J}\left(Jt + \sin(Jt)\right) \ , \\
C_1(t) &= i\frac{Jt}{8} + \frac{h}{4J}\left(1 - \cos(Jt)\right) \ , \\
C_2(t) &= -i\frac{h}{4J}\left(Jt - \sin(Jt)\right) \ .
\end{aligned} \quad (19)$$

Analogously for $d = 2$,

$$\begin{aligned}
\mathscr{H}_{2D}^{(1)} = \sum_l \Bigg[ & C_0^{(1)}(t) + C_1^{(1)}(t)\sum_{a\in\mathscr{V}_1^l} s_a^z s_l^z + C_2^{(1)}(t)\sum_{(a,b)\in\mathscr{V}_2^l} s_a^z s_b^z \\
& + C_3^{(1)}(t)\sum_{(a,b,c)\in\mathscr{V}_4^l} s_a^z s_b^z s_c^z s_l^z + C_4^{(1)}(t)\sum_{(a,b,c,d)\in\mathscr{V}_3^l} s_a^z s_b^z s_c^z s_d^z \Bigg],
\end{aligned} \quad (20)$$

where

$$\begin{aligned}
C_0^{(1)}(t) &= i\frac{h}{2J}\frac{6Jt + 8\sin(Jt) + \sin(2Jt)}{16} \ , \\
C_1^{(1)}(t) &= i\frac{Jt}{8} + \frac{h}{2J}\frac{1 - \cos^4(Jt/2)}{2J} \ , \\
C_2^{(1)}(t) &= -i\frac{h}{2J}\frac{2Jt - \sin(2Jt)}{16} \ , \\
C_3^{(1)}(t) &= -\frac{h}{2J}\frac{\sin^4(Jt/2)}{2J} \ , \\
C_4^{(1)}(t) &= i\frac{h}{2J}\frac{6Jt - 8\sin(Jt) + \sin(2Jt)}{16} \ .
\end{aligned} \quad (21)$$

The classical network from first-order cumulant expansion in $d = 3$ is given by

$$
\begin{aligned}
\mathcal{H}_{3D}^{(1)} = \sum_l \Bigg[ & C_0^{(1)}(t) + C_1^{(1)}(t) \sum_{a \in \mathcal{V}_1^l} s_a^z s_l^z + C_2^{(1)}(t) \sum_{(a,b) \in \mathcal{V}_2^l} s_a^z s_b^z \\
& + C_3^{(1)}(t) \sum_{(a,b,c) \in \mathcal{V}_3^l} s_a^z s_b^z s_c^z s_l^z + C_4^{(1)}(t) \sum_{(a,b,c,d) \in \mathcal{V}_4^l} s_a^z s_b^z s_c^z s_d^z \\
& + C_5^{(1)}(t) \sum_{(a,b,c,d,e) \in \mathcal{V}_5^l} s_a^z s_b^z s_c^z s_d^z s_e^z s_l^z + C_6^{(1)}(t) \sum_{(a,b,c,d,e,f) \in \mathcal{V}_6^l} s_a^z s_b^z s_c^z s_d^z s_e^z s_f^z \Bigg], \quad (22)
\end{aligned}
$$

with

$$
\begin{aligned}
C_0^{(1)}(t) &= i \frac{h}{2J} \frac{30Jt + 45\sin(Jt) + 9\sin(2Jt) + \sin(3Jt)}{96}, \\
C_1^{(1)}(t) &= i \frac{Jt}{8} + \frac{h}{2J} \frac{1 - \cos^6(Jt/2)}{3}, \\
C_2^{(1)}(t) &= -i \frac{h}{2J} \frac{6Jt + 3\sin(Jt) - 3\sin(2Jt) - \sin(3Jt)}{96}, \\
C_3^{(1)}(t) &= -\frac{h}{2J} \frac{\sin^4(Jt/2)(\cos(Jt) + 2)}{6}, \\
C_4^{(1)}(t) &= i \frac{h}{2J} \frac{6Jt - 3\sin(Jt) - 3\sin(2Jt) + \sin(3Jt)}{96}, \\
C_5^{(1)}(t) &= \frac{h}{2J} \frac{\sin^6(Jt/2)}{3}, \\
C_6^{(1)}(t) &= -i \frac{h}{2J} \frac{30Jt - 45\sin(Jt) + 9\sin(2Jt) - \sin(3Jt)}{96}.
\end{aligned} \quad (23)
$$

## A.2 Range of applicability and effect of higher order terms

Fig. 6 shows the time evolution of transverse magnetization and nearest-neighbor spin-spin correlation obtained from the first-order cumulant expansion for different $h/J$. We find that for $ht < 1$ the results from the cumulant expansion agree with the exact results to a similar extent independent of the value of $h/J$. For $ht > 1$ the cumulant expansion deviates strongly from the exact results.

To second order in the cumulant expansion the wave function coefficients are approximated by

$$
\begin{aligned}
\frac{\psi(\vec{s}, t)}{\psi_0(\vec{s})} &= \frac{\langle \vec{s}| e^{-iHt} |\psi_0\rangle}{\langle \vec{s}|\psi_0\rangle} \\
&\approx e^{-iE_{\vec{s}}t} \exp\Bigg[ -i\lambda \int_0^t dt' \frac{\langle \vec{s}|V(t')|\psi_0\rangle}{\langle \vec{s}|\psi_0\rangle} \\
&\quad - \lambda^2 \int_0^t dt' \int_0^{t'} dt'' \left( \frac{\langle \vec{s}|V(t')V(t'')|\psi_0\rangle}{\langle \vec{s}|\psi_0\rangle} - \frac{\langle \vec{s}|V(t')|\psi_0\rangle \langle \vec{s}|V(t'')|\psi_0\rangle}{\langle \vec{s}|\psi_0\rangle^2} \right) \Bigg]. \quad (24)
\end{aligned}
$$

In one dimension this yields the effective Hamilton function of the general form

$$
\mathcal{H}^{(2)}(\vec{s}, t) = \sum_{n_1=0}^z \sum_{n_2=0}^z C_{n_1 n_2}(t) \sum_{l=1}^N \sum_{(a_1,\ldots,a_{n_1}) \in \mathcal{V}_{n_1}^{1l}} \sum_{(b_1,\ldots,b_{n_2}) \in \mathcal{V}_{n_2}^{2l}} s_l^{n_1+n_2} \prod_{r_1=1}^{n_1} s_{a_{r_1}} \prod_{r_2=1}^{n_2} s_{b_{r_2}}, \quad (25)
$$

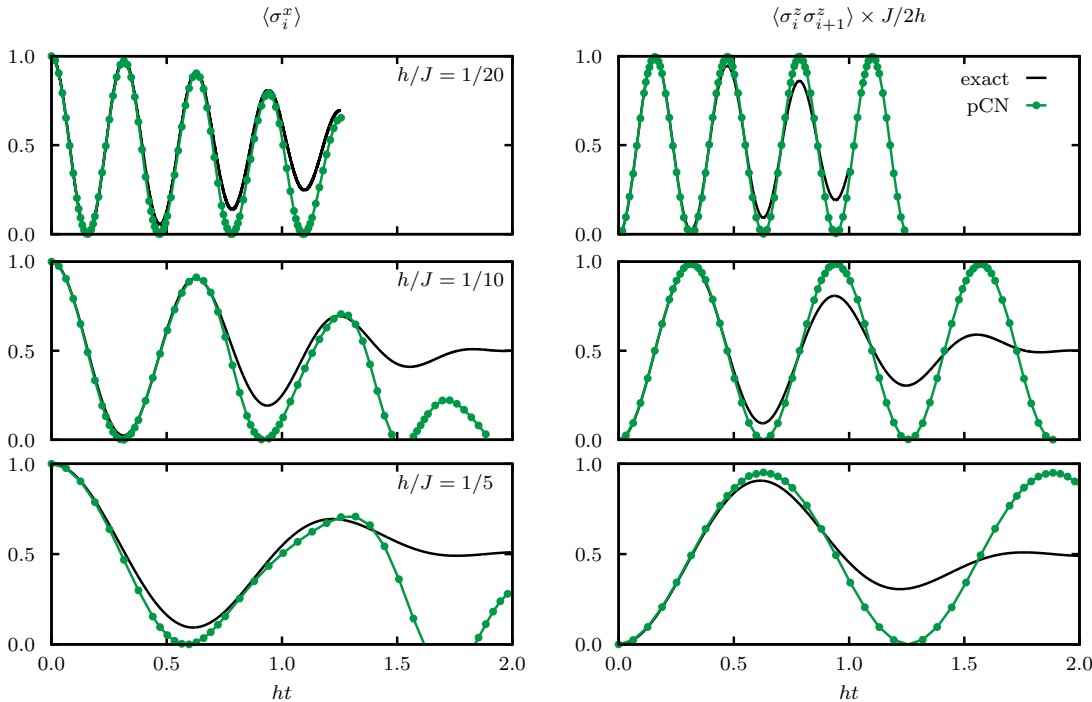

Figure 6: MC data from pCN in comparison with exact results for different $h/J$ in $d = 1$. The left column shows magnetization $\langle \sigma_i^x \rangle$ and the right column shows spin-spin correlation $\langle \sigma_i^z \sigma_{i+1}^z \rangle$.

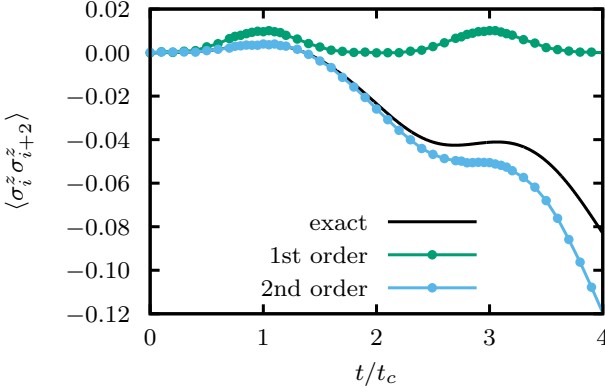

Figure 7: Next-nearest-neighbor correlation function in $d = 1$ obtained with first-order and second-order cumulant expansion in comparison with the exact result; $h/J = 0.05$.

where $\mathcal{V}_n^{dl}$ denotes the set of all groups of $n$ spins at distance $d$ from spin $l$. The coupling constants are

$$C_{00}(t) = i\frac{h}{4J}\left(Jt + \sin(Jt)\right) - \frac{h^2}{J^2}\sin(Jt/2) \, ,$$

$$C_{10}(t) = i\frac{Jt}{8} + \frac{h}{4J}\left(1 - \cos(Jt)\right) + i\frac{h^2}{8J^2}\left(2Jt - 4\sin(Jt) + \sin(2Jt)\right) \, ,$$

$$C_{20}(t) = -i\frac{h}{4J}\left(Jt - \sin(Jt)\right) - \frac{h^2}{J^2}\sin(Jt/2) \, ,$$

$$C_{01}(t) = \frac{h^2}{32J^2}\Big(9 - 2J^2t^2 - 8\cos(Jt) - \cos(2Jt) - 4Jt\sin(Jt)\Big),$$

$$C_{11}(t) = i\frac{h^2}{32J^2}\Big(6Jt - 8Jt\cos(Jt) + \sin(2Jt)\Big),$$

$$C_{21}(t) = \frac{h^2}{16J^2}\Big(\sin(Jt) - Jt\Big)^2,$$

$$C_{02}(t) = 0, \quad C_{12}(t) = 0, \quad C_{22}(t) = 0. \tag{26}$$

We observe that taking into account the second order contribution of the cumulant expansion significantly enhances the result for the next-nearest-neighbor correlation function as shown in Fig. A. In particular it yields corrections that are much larger than what one would expect from a naive perturbative expansion.

### A.3   Comparison: Complexity of the equivalent iMPS

In order to give an estimate of the complexity of the time-evolved state in terms of Matrix Product States we show the time evolution of local observables, entanglement, and bond dimension after the quench $h_0 = \infty \rightarrow h = J/20$ computed using iTEBD [77] in Fig. 8. The bond dimension $\chi$ (i.e. the number of singular values kept after singular value decompositions) was restricted to different maximal values $\chi_{max}$ and during the simulation Schmidt values smaller than $10^{-10}$ were discarded. In all quantities a converged result on the time interval of interest is obtained with a maximal bond dimension of $\chi_{max} \geq 4$.

For the implementation of the iTEBD algorithm the iTensor library [76] was used.

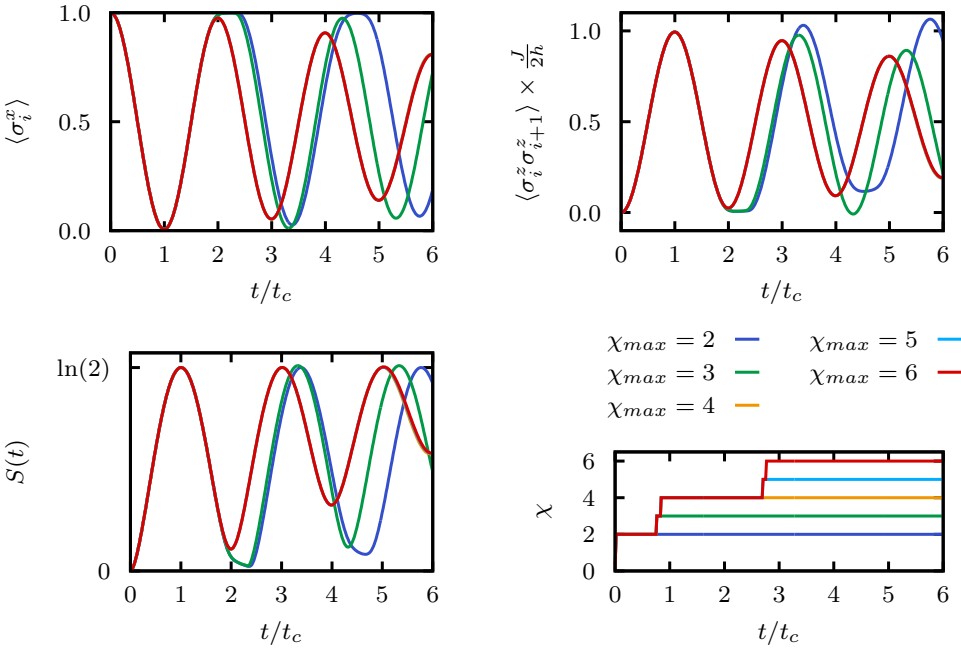

Figure 8: Dynamics for the quench from $h_0 = \infty$ to $h/J = 0.05$ computed with iTEBD with different maximal bond dimensions $\chi_{max}$.

# B  Loschmidt amplitude as classical partition function

## B.1  Real weights from decimation RG

As outlined in the results section the Loschmidt amplitude (8) after integrating out every second spin, residing on sublattice $\Lambda$, can be integrated out, yielding

$$Z(t) = \frac{1}{2^N} \sum_{\vec{s} \in \{\pm 1\}^{N/2}} \prod_{i \in \Lambda} 2\cos\left( \frac{J}{4} t \sum_{\langle i,j \rangle} s_j \right). \tag{27}$$

A Hamilton function $\mathscr{H}(\vec{s}, t)$ defining a classical network can be obtained by choosing a general ansatz including all possible $\mathbb{Z}_2$-symmetric couplings of spins with a common neighbor on the sublattice $\Lambda$, which takes the form given in Eq. (10). The Boltzmann weight of a configuration is then given by

$$e^{\mathscr{H}(\vec{s},t)} = \prod_{l \in \Lambda} \exp\left[ \sum_{n=0}^{z/2} C_n(t) \sum_{(a_1,\ldots,a_{2n}) \in \mathscr{V}_{2n}^l} \prod_{r=1}^{2n} s_{a_r} \right]. \tag{28}$$

Equating each factor in the expression above with the corresponding factor in Eq. (27) for every configuration of the involved spins yields a system of equations that determines the couplings $C_n(t)$ [65].

In $d = 1$ the couplings are

$$C_0(t) = \ln 2 + \frac{\ln\left( \cos(Jt/2) \right)}{2} , \quad C_1(t) = \frac{\ln\left( \cos(Jt/2) \right)}{2} . \tag{29}$$

The couplings in $d = 2$ are

$$
\begin{aligned}
C_0(t) &= \ln 2 + \frac{\ln\left( \cos(Jt) \right) + 4\ln\left( \cos(Jt/2) \right)}{8} , \\
C_1(t) &= \frac{\ln\left( \cos(Jt) \right)}{8} , \\
C_2(t) &= \frac{\ln\left( \cos(Jt) \right) - 4\ln\left( \cos(Jt/2) \right)}{8} .
\end{aligned}
\tag{30}
$$

In $d = 3$ the resulting couplings are

$$
\begin{aligned}
C_0(t) &= \ln 2 + \frac{\ln\left( \cos(3Jt/2) \right) + 6\ln\left( \cos(Jt) \right) + 15\ln\left( \cos(Jt/2) \right)}{32} , \\
C_1(t) &= \frac{\ln\left( \cos(3Jt/2) \right) + 2\ln\left( \cos(Jt) \right) - \ln\left( \cos(Jt/2) \right)}{32} , \\
C_2(t) &= \frac{\ln\left( \cos(3Jt/2) \right) - 2\ln\left( \cos(Jt) \right) - \ln\left( \cos(Jt/2) \right)}{32} , \\
C_3(t) &= \frac{\ln\left( \cos(3Jt/2) \right) - 6\ln\left( \cos(Jt) \right) + 15\ln\left( \cos(Jt/2) \right)}{32} .
\end{aligned}
\tag{31}
$$

The time evolution of these couplings is displayed in Fig. 9.

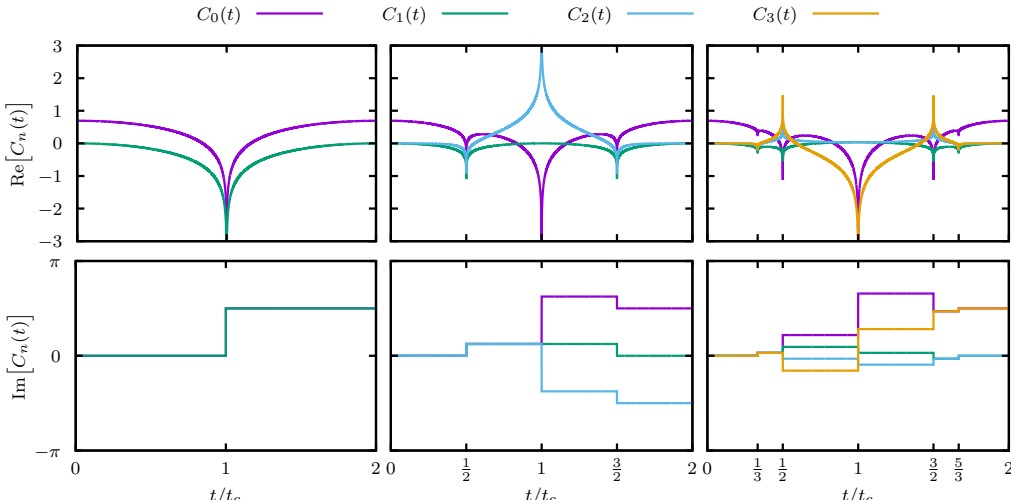

Figure 9: Time evolution of the couplings of the effective Hamilton function $\mathscr{H}(\vec{s}, t)$ for the Loschmidt amplitude in one, two, and three dimensions.

## B.2 Monte-Carlo scheme for the Loschmidt amplitude

In order to evaluate the Loschmidt amplitude given in terms of the renormalized Boltzmann weights (28) a combination of different Monte Carlo techniques is employed. Since the Loschmidt amplitude is the normalization of the Boltzmann weights a simple Metropolis Monte Carlo sampling is not sufficient. Moreover, the Monte Carlo sampling is hindered by critical slowing down close to the critical times and the presence of negative weights leads to a sign problem.

The idea to deal with these issues is to sample for a given Hamilton function $\mathscr{H}(\vec{s}, t)$ the energy histograms $P_{\pm}(E) = \Omega_{\pm}(E)e^{E}$ where the density of states $\Omega_{\pm}(E)$ is the number of configurations $\vec{s}$ with energy $E = \operatorname{Re}\mathscr{H}(\vec{s}, t)$. The sign index indicates the sign of the corresponding Boltzmann weight. Given a good estimate of these histograms the partition sum is simply

$$Z(t) = \sum_{E, \sigma = \pm 1} \sigma \, P_{\sigma}(E) \,. \tag{32}$$

Note, however, that the histograms $P_{\pm}(E)$ must be properly normalized in order to get the correct result for $Z(t)$. In order to obtain a good estimate of the normalized histogram we combine the following techniques:

1. *Separate sampling of factor graphs.* In order to overcome the sign problem the configuration space $\mathscr{X} = \{\pm 1\}^{N'}$ is separated into $\mathscr{X}_{+} = \{\vec{s}|e^{\mathscr{H}(\vec{s}, t)} > 0\}$ and $\mathscr{X}_{-} = \{\vec{s}|e^{\mathscr{H}(\vec{s}, t)} < 0\}$; $N'$ is the number renormalized spins. Then the partition sum is split as

$$
\begin{aligned}
Z(t) &= Z_{+}(t) + Z_{-}(t), \\
Z_{\pm} &= \sum_{\vec{s} \in \mathscr{X}_{\pm}} e^{\mathscr{H}(\vec{s}, t)} = \pm \sum_{E} P_{\pm}(E) \,.
\end{aligned}
\tag{33}
$$

The partition sums $Z_{\pm}$ can be sampled separately as described in Ref. [67].

2. *Importance sampling.* When sampling the energy $E$ in an importance sampling scheme with weights $e^{E}$ the relative frequency of samples with energy $E$ is proportional to $P_{\pm}(E) = \Omega_{\pm}(E)e^{E}$. Therefore, a histogram of the energies sampled with Metropolis Monte Carlo updates yields the desired histograms up to normalization. Moreover, the importance sampling allows to choose the region in the energy spectrum that is sampled by introducing an artificial temperature as described next.

3. *Parallel tempering.* Parallel tempering [68] is a method to improve the sampling efficiency in strongly peaked multi-modal distributions, which occurs in our case close to the critical times. The idea of parallel tempering is to perform a Markov Chain Monte-Carlo (MCMC) sampling on several copies of a system at different temperatures. During the sampling the system configurations are not only updated as usual but also configuration swaps between adjacent temperatures are possible. Thereby a MCMC on the temperatures is performed allowing the system to jump between different peaks of the distribution.

In the present case a distribution with weights $w(\vec{s}, t) = e^{\mathcal{H}(\vec{s}, t)}$ shall be sampled. Introducing an artificial temperature $\beta$ yields weights

$$w_\beta(\vec{s}, t) = e^{\beta \mathcal{H}(\vec{s}, t)} . \tag{34}$$

At $\beta = 1$ the sampling is inefficient due to the diverging renormalized weights of the Hamilton function (see bottom panels in Fig. 4). This problem is attenuated if we sample with a parallel tempering scheme with temperatures $1 = \beta_1 > \beta_2 > \ldots > \beta_N$. Moreover, parallel tempering is beneficial, because histograms $P_\pm^\beta(E) = \Omega_\pm(E) e^{\beta E}$ are obtained as a byproduct, which capture different regions of the spectrum with high precision. This can be used to obtain decent precision over the whole range of energies and thereby a properly normalized histogram as described next.

4. *Multiple histogram reweighting.* In order to get a good histogram for $P_\pm(E)$ in the whole energy range the fact that

$$P_\pm^{\beta_1}(E) = e^{(\beta_1 - \beta_0)E} P_\pm^{\beta_0}(E) \tag{35}$$

can be expoited. In the multiple histogram reweighting procedure [69] the histograms obtained at the different temperatures are combined to yield a histogram covering the whole energy range. This allows us to normalize the histogram at $\beta = 0$, where

$$\sum_{E, \sigma = \pm 1} |P_\sigma^{\beta = 0}(E)| = 2^{N'} . \tag{36}$$

## B.3 Simplification of effective systems close to $t_c$

For times $t$ close to the critical time $t_c$ the effective classical networks can be simplified, because some of the couplings become very small, as evident from Fig. 4 and also Fig. 9, and the Hamilton function is dominated by the divergent contributions. This simplification can be exploited for additional insights into the behavior of the Loschmidt amplitude close to the critical time. In the following we will discuss the case $d = 2$, but the arguments hold similarly for $d = 3$.

Dropping contributions to the couplings that vanish at $t_c$ the partition sum close to $t_c$ can be approximated by

$$Z(t) \approx \frac{1}{2^{N'}} \sum_{\vec{s} \in \{\pm 1\}^{N'}} \sigma_{\vec{s}} \, e^{-\beta(t)\bar{\mathcal{H}}(\vec{s})}, \tag{37}$$

with an effective temperature $\beta(t) = -\ln\big(\cos(Jt/2)\big)/2$, the number of remaining spins $N' = N/2$, $\sigma_{\vec{s}} = \pm 1$ the sign of the weight of the configuration $\vec{s}$, and

$$\bar{\mathcal{H}}(\vec{s}) = \sum_{i,j} \big(1 - s_{i,j}s_{i+1,j}s_{i,j+1}s_{i+1,j+1}\big) . \tag{38}$$

The minimal energy of the network defined by $\bar{\mathcal{H}}(\vec{s})$ is obviously reached when the condition

$$s_{i,j}s_{i+1,j}s_{i,j+1}s_{i+1,j+1} = 1 \tag{39}$$

is fulfilled on each plaquette. This is possible in systems where the edge lengths of the system, $N_x'$ and $N_y'$, are both even, to which we restrict the following discussion. To obtain a "ground state" it is sufficient to fix the spin configuration in one row and in one column. The state of the remaining spins is then determined by the condition (39). Hence, the ground state is $2^{N_x'+N_y'-1}$-fold degenerate.

From Eq. (27) we know that the sign of the corresponding Boltzmann weight is determined by the number of plaquettes with $|s_{i,j}+s_{i+1,j}+s_{i,j+1}+s_{i+1,j+1}| = 4$. If there is an even number of plaquettes with this property, the configuration has a positive Boltzmann weight, otherwise it is negative. We find that for even edge lengths the ground states always have positive Boltzmann weights.

Let us now introduce the density of states $\Omega_{\pm}(E)$, i.e. the number of spin configurations $\vec{s}$ with the same real part of the energy $E = \mathcal{H}(\vec{s},t)$ and $\text{sgn}(e^{\mathcal{H}(\vec{s},t)}) = \pm 1$, in order to rewrite the sum over configurations in Eq. (37) as a sum over energies,

$$Z(t) = \frac{1}{2^{N'}} \sum_{E,\sigma=\pm 1} \sigma \Omega_{\sigma}(E) e^{-\beta(t)E} . \tag{40}$$

From the above analysis of the ground state we know that $\Omega_+(0) = 2^{N_x'+N_y'-1}$. In the limit $t \to t_c$, or equivalently $\beta \to \infty$, this is the only contribution that does not vanish in the sum. Therefore, $Z(t_c) = 2^{N_x'+N_y'-1-N'}$ and

$$\lambda_N(t_c) = \left( \frac{1}{2} - \frac{N_x'+N_y'-1}{N} \right) \ln 2 \xrightarrow{N \to \infty} \frac{\ln 2}{2} , \tag{41}$$

which determines the value of the rate function at $t_c$ in the thermodynamic limit and the finite size correction.

We would like to remark that classical spin systems of the form (38) were studied in the literature and can be solved analytically for real temperatures [78, 79]. We found, however, that introducing a sign into the partition sum renders the analytical summation impossible.

## C  Exemplary derivation of ANN couplings from the cumulant expansion

### C.1  $d = 1$

From the cumulant expansion (18) we have

$$\mathscr{P}_l(\vec{s},t) = C_0(t) + C_1(t)s_l(s_{l-1}+s_{l+1}) + C_2(t)s_{l-1}s_{l+1} , \tag{42}$$

i.e.

$$\psi(\vec{s}) = \prod_l \exp\left( C_0(t) + C_1(t)s_l(s_{l-1}+s_{l+1}) + C_2(t)s_{l-1}s_{l+1} \right) . \tag{43}$$

A patch consists of three consecutive spins and swapping the two spins at the border leaves the weight unchanged.

A possible ansatz for the ANN with one hidden spin per lattice site (see Fig. 5(a) of the main text), that respects the symmetries, is

$$\psi(\vec{s}) = \left( \frac{\Omega}{2} \right)^N \sum_{\vec{u}^{(1)},\vec{u}^{(2)}} \exp\left( \sum_l \left( W_1(s_{l-1}+s_{l+1}) + W_2 s_l \right) u_l \right) , \tag{44}$$

where $\Omega$ constitutes a overall normalization and phase that is irrelevant when expectation values are computed with the Metropolis algorithm. Integrating out the hidden spins yields

$$\psi(\vec{s}) = \prod_l \Omega \cosh\left(W_1(s_{l-1} + s_{l+1}) + W_2 s_l\right) \tag{45}$$

Identifying the single factors yields for the different possible spin configurations (in the following we abbreviate cosh by ch)

$$
\begin{aligned}
\uparrow\uparrow\uparrow: &\quad \Omega\,\mathrm{ch}(2W_1 + W_2) = \exp(C_0 + 2C_1 + C_2) \\
\uparrow\uparrow\downarrow: &\quad \Omega\,\mathrm{ch}(W_2) = \exp(C_0 - C_2) \\
\uparrow\downarrow\uparrow: &\quad \Omega\,\mathrm{ch}(2W_1 - W_2) = \exp(C_0 - 2C_1 + C_2)
\end{aligned}
\tag{46}
$$

All other spin configurations are connected to these via $\mathbb{Z}_2$ symmetry. This is an implicit equation for the ANN weights that can be solved numerically. One solution for the weights obtained from the 1st order cumulant expansion is plotted in Fig. 5(b) of the main text. Note that these equations have different possible solutions.

## C.2  $d = 2$

From the cumulant expansion (20) we have

$$
\begin{aligned}
\mathscr{P}_l(\vec{s}, t) ={}& C_0^{(1)}(t) + C_1^{(1)}(t) \sum_{a \in \mathscr{V}_1^l} s_a^z s_l^z + C_2^{(1)}(t) \sum_{(a,b) \in \mathscr{V}_2^l} s_a^z s_b^z \\
&+ C_3^{(1)}(t) \sum_{(a,b,c) \in \mathscr{V}_3^l} s_a^z s_b^z s_c^z s_l^z + C_4^{(1)}(t) \sum_{(a,b,c,d) \in \mathscr{V}_4^l} s_a^z s_b^z s_c^z s_d^z \,,
\end{aligned}
\tag{47}
$$

A patch consists of a central spin $s_{i,j}$ and four neighboring spins as depicted by the black dots in Fig. 4a in the main text. Any permutation of the surrounding spins leaves $\mathscr{P}_l(\vec{s}, t)$ unchanged.

A possible ansatz for the ANN with five hidden spins per lattice site is depicted in Fig. 5(c) of the main text. After integrating out the hidden spins the wave function is given by

$$
\begin{aligned}
\psi(\vec{s}) = \Omega \prod_l &\,\mathrm{ch}\left(W^{(1)} s_{i,j}\right)\mathrm{ch}\left(W_1^{(1)} s_{i,j} + W_2^{(1)}(s_{i,j+1} + s_{i,j-1} + s_{i+1,j} + s_{i-1,j})\right) \\
&\times \mathrm{ch}\left(W_1^{(2)} s_{i,j} + W_2^{(2)}(s_{i,j+1} + s_{i,j-1} + s_{i+1,j})\right) \\
&\times \mathrm{ch}\left(W_1^{(2)} s_{i,j} + W_2^{(2)}(s_{i,j+1} + s_{i,j-1} + s_{i-1,j})\right) \\
&\times \mathrm{ch}\left(W_1^{(2)} s_{i,j} + W_2^{(2)}(s_{i+1,j} + s_{i-1,j} + s_{i,j+1})\right) \\
&\times \mathrm{ch}\left(W_1^{(2)} s_{i,j} + W_2^{(2)}(s_{i+1,j} + s_{i-1,j} + s_{i,j-1})\right).
\end{aligned}
\tag{48}
$$

Identifying the single factors yields for the different possible spin configurations

$$
\begin{aligned}
\uparrow\uparrow\uparrow\uparrow\uparrow: &\quad \Omega\,\mathrm{ch}\left(W_1^{(1)} + 4W_2^{(1)}\right)\mathrm{ch}\left(W_1^{(2)} + 3W_2^{(2)}\right)^4 \\
&= \exp\left(4C_1 + 4C_3 + C_0 + 6C_2 + C_4\right) \\
\uparrow\uparrow\uparrow\uparrow\downarrow: &\quad \Omega\,\mathrm{ch}\left(W_1^{(1)} + 2W_2^{(1)}\right)\mathrm{ch}\left(W_1^{(2)} + 3W_2^{(2)}\right)\mathrm{ch}\left(W_1^{(2)} + W_2^{(2)}\right)^3 \\
&= \exp\left(2C_1 - 2C_3 + C_0 - C_4\right) \\
\uparrow\uparrow\uparrow\downarrow\downarrow: &\quad \Omega\,\mathrm{ch}\left(W_1^{(1)}\right)\mathrm{ch}\left(W_1^{(2)} + W_2^{(2)}\right)^2\mathrm{ch}\left(W_1^{(2)} - W_2^{(2)}\right)^2 \\
&= \exp\left(C_0 - 2C_2 + C_4\right) \\
\downarrow\uparrow\uparrow\uparrow\uparrow: &\quad \Omega\,\mathrm{ch}\left(-W_1^{(1)} + 4W_2^{(1)}\right)\mathrm{ch}\left(-W_1^{(2)} + 3W_2^{(2)}\right)^4
\end{aligned}
$$

$$= \exp\left(-4C_1 - 4C_3 + C_0 + 6C_2 + C_4\right)$$

$$\downarrow\uparrow\uparrow\uparrow\downarrow: \quad \Omega \ \mathrm{ch}\left(-W_1^{(1)} + 2W_2^{(1)}\right)\mathrm{ch}\left(-W_1^{(2)} + 3W_2^{(2)}\right)\mathrm{ch}\left(-W_1^{(2)} + W_2^{(2)}\right)^3$$

$$= \exp\left(-2C_1 + 2C_3 + C_0 - C_4\right) \tag{49}$$

where the leftmost arrow in the spin configurations corresponds to the central spin of the patch. One solution for the weights obtained from the 1st order cumulant expansion is plotted in Fig. 5(d) of the main text.

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
