# Peer review of "Quantum dynamics in transverse-field Ising models from classical networks"

_SciPost Physics, doi:SciPost Phys. 4, 013 (2018)_

## Round 3 · Referee Report · Anonymous · 2017-12-16

Strengths

1) Introduces a new ansatz based on classical networks to obtain dynamics of quantum many body systems
2) This approach is more accurate than usual perturbation theory
3) The method works in any dimension and is not restricted to 1D or quasi-2D, as MPS/DMRG methods are.
4) Getting the classical network can help to develop neural network ansatzes for the quantum dynamics of generic many body systems, which recently has been found to work reliably in 2D Ising systems (see Ref. [23] in the manuscript).

Weaknesses

1) The results compare well with the exact solution only at small times.
2) As an outlook, the possibility to further develop the method is described, but it is difficult to estimate how well this might work.

Report

The authors introduce a method to compute the time evolution of quantum many body systems via a mapping to classical networks, whose properties are then evaluated using classical Monte Carlo techniques. The results presented are obtained for the transverse field Ising model in 1D, 2D and 3D. Local observables as well as (short-range) correlations are computed, as well as the entanglement entropy and the rate function of the Loschmidt amplitude. The latter allows one to investigate for dynamical quantum phase transitions, which occur as non-analytic behavior in the course of time when going to the thermodynamic limit.
The approach is a very interesting idea and opens the path to further developments. As the authors show, it works reliably at short times, but looses its accuracy on time scales smaller than the typical ones accessible to MPS methods in 1D. However, the approach is more flexible than MPS since it is not restricted to low-D systems. In addition, the results shown most probably have room for improvement by going to higher orders of the cumulant expansion used, or by using other techniques for evaluating the networks. Already on the level presented, the results are more accurate than perturbation theory, which is rather encouraging. The manuscript is well written and I think is ready for publication in SciPost Physics after including the following minor changes:

- Figure 1a) is hard to understand and needs better explanation, either in a legend, or in a much more detailed caption.

- The authors report that in their classical Monte Carlo approach, negative probabilities appear. This sign problem is apparently resolved using the approach introduced in Ref. [61] of the manuscript. However, this is only briefly mentioned in the main text on page 7 and then discussed in some detail in the appendix. I would find it useful to say more about the significance of this problem for the calculations and its solution in the main text.

- I find Sec. 2.5 a little confusing: apparently, a mapping to an artificial neural network (ANN) was done and the time evolutions of the couplings are discussed in Fig. 5, but no results for observables is presented. It is not clear to me if there is a deeper reason for this, or whether the results obtained by an ANN will be identical to the ones already shown before.

- In Fig. 6, probably the black line shows the exact results; for the sake of clarity I suggest to add a legend saying this and to describe the different colors used, or at least mention in the caption what the various colors mean.

- In Appendix A.3, as complementary information the authors discuss the complexity of an equivalent iTEBD calculation. In the Conclusions section, this is referred to, but it is not really discussed how the computational complexity of both ansatzes compares. To be more specific, the authors say that the network ansatz in first order keeps only three couplings, while the iMPS needs 64. However, this does not automatically imply that the approach with the fewer couplings needs smaller computational resources, and one needs to keep in mind, that a good accuracy is obtained only at small times when using the networks. I would find it useful to comment or compare in the appendix the computational resources (e.g., CPU-time, memory) needed to reach a faithful result at short times (e.g., times < 2) using both approaches.

Requested changes

1) improve Fig. 1a and caption
2) better discuss negative probabilities in the main text
3) results for observables using ANNs?
4) improve Fig. 6
5) comment and/or compare resources needed for iMPS and the network ansatzes.

---

## Round 3 · Referee Report · Anonymous · 2017-12-25

Strengths

1- Presents a systematic method, which can be used to study quantum dynamics in a general system.
2- The method can be used also in higher dimensional systems.
3- The method works - in principle - for general quantum systems having short-range interactions.
4- The method provides accurate numerical results for short times.

Weaknesses

1- The accuracy of the method is limited to short times.
2- Higher order (3rd, 4th, etc.) expansions seem to be very cumbersome.

Report

In this paper the quantum dynamics of transverse-field Ising models is studied in a formalism of classical networks, which is evaluated by standard Monte-Carlo methods. The method, based on the cumulant expansion seems to be quite general, which can - in principle - be used also in higher dimensions and can be generalised by other quantum systems with short-range interactions. Using this method the authors have calculated the time-dependence of local observables, near-neighbour correlations, two-site entanglement and the rate function of the Loschmidt echo. The results are accurate for short times and in principle can be improved by considering higher order terms of the expansion.

The subject of the paper is of interest of researchers is quantum physics. It is basically well written and merit publication in the SciPost, however some points needs to be clarified.

i) The presented examples in the text are for a small perturbation term: $h/J=0.05$ and mainly in first order of the cumulant expansion. The authors should try to estimate the time-window in which the higher order expansion terms are still accurate.

ii) In two-dimensions the transverse-field Ising model has been studied before in Refs.[20,21]. The authors should try to compare their numerical results with the previous one, if possible.

iii) The results in the paper are about local observables, near-neighbour correlations and two-site entanglement. Can the method also be used to calculate long-range correlations, block-entanglement, etc.?

iv) Fig. 1a appears quite early and needs more explanation to be understandable.

v) The quantity "maximal bond dimension" should be defined in context with the iMPS.

vi) For the transverse-field Ising model in 1d the time-evolution after a quench of several observables in the free-fermion method have been first calculated in Phys. Rev. A 2, 1075 (1970); Phys. Rev. A 3, 786 (1971); Phys. Rev. A 3, 2137 (1971); Phys. Rev. Lett. 85, 3233 (2000) and Phys. Rev. A 69, 053616 (2004). These papers are recommended to be cited.

Requested changes

1- Fig. 1a appears quite early and needs more explanation.
2- The quantity "maximal bond dimension" should be defined in context with the iMPS.
3- For the transverse-field Ising model in 1d the time-evolution after a quench of several observables in the free-fermion method have been first calculated in Phys. Rev. A 2, 1075 (1970); Phys. Rev. A 3, 786 (1971); Phys. Rev. A 3, 2137 (1971); Phys. Rev. Lett. 85, 3233 (2000) and Phys. Rev. A 69, 053616 (2004). These papers are recommended to be cited.

---

## Round 4 · Referee Report · Anonymous · 2018-2-13

Report

The authors have successfully modified the manuscript, I suggest publication in the present form.

---

## Round 4 · Author Response

We thank the Referees for their helpful and supportive comments regarding our manuscript. We have considered all suggestions in the preparation of our revised manuscript. A detailed list of the changes in our manuscript can be found in the response to the Referees below.

REPORT 1:

1)

-- Referee: "Figure 1a) is hard to understand and needs better explanation, either in a legend, or in a much more detailed caption."

-- Response: We thank the Referee for pointing this out. We expanded significantly the explanation in the caption of Fig. 1, in order to make the content of the figure more easily accessible.

2)

-- Referee: "The authors report that in their classical Monte Carlo approach, negative probabilities appear. This sign problem is apparently resolved using the approach introduced in Ref. [61] of the manuscript. However, this is only briefly mentioned in the main text on page 7 and then discussed in some detail in the appendix. I would find it useful to say more about the significance of this problem for the calculations and its solution in the main text."

-- Response: We agree that the treatment of negative weights merits extended discussion in the main text. Accordingly, we expanded the explanation below Eq. (10), where we now discuss how it is possible to sample the complex partition function even in the presence of complex weights.

3)

-- Referee: "I find Sec. 2.5 a little confusing: apparently, a mapping to an artificial neural network (ANN) was done and the time evolutions of the couplings are discussed in Fig. 5, but no results for observables is presented. It is not clear to me if there is a deeper reason for this, or whether the results obtained by an ANN will be identical to the ones already shown before."

-- Response: The mapping from our pCNs to ANNs is exact, which means that observables sampled using the ANN wavefunction are automatically identical with those from the pCN wavefunction. We anticipate, however, that this might not have been clear from the prior presentation. Therefore, we added a sentence above Eq. (11) to clarify this.

4)

-- Referee: "In Fig. 6, probably the black line shows the exact results; for the sake of clarity I suggest to add a legend saying this and to describe the different colors used, or at least mention in the caption what the various colors mean."

-- Response: We added a legend to Fig. 6 and refrained from using different colors to avoid confusion.

5)

-- Referee: "In Appendix A.3, as complementary information the authors discuss the complexity of an equivalent iTEBD calculation. In the Conclusions section, this is referred to, but it is not really discussed how the computational complexity of both ansatzes compares. To be more specific, the authors say that the network ansatz in first order keeps only three couplings, while the iMPS needs 64. However, this does not automatically imply that the approach with the fewer couplings needs smaller computational resources, and one needs to keep in mind, that a good accuracy is obtained only at small times when using the networks. I would find it useful to comment or compare in the appendix the computational resources (e.g., CPU-time, memory) needed to reach a faithful result at short times (e.g., times < 2) using both approaches."

-- Response: We agree with the Referee, that a quantitative assessment of the computational complexity using our perturbative approach compared to iMPS is an interesting and also important aspect. For such an analysis it would be most important to study the scaling behavior of numerical resources at long evolution times. This late-time regime, however, we don’t yet attempt to describe within the pCN in its current stage, such that a comparison of the computational complexity we currently cannot achieve. We see, however, from the Referee’s comment that we might have indicated in our manuscript that we have performed such a complexity comparison. In order to avoid this impression, we removed the corresponding comment from the Conclusions section.

REPORT 2:

i)

-- Referee: "The presented examples in the text are for a small perturbation term: $h/J=0.05$and mainly in first order of the cumulant expansion. The authors should try to estimate the time-window in which the higher order expansion terms are still accurate."

-- Response: We thank the Referee for this comment. For the data shown in the main text of the manuscript, we have chosen for consistency one particular ratio h/J which is well within the considered perturbative limit. Further data for different values of $h/J$ are included in Appendix A.2. We anticipate, however, that it might not have been evident from the previous manuscript, that we also performed a quantitative analysis for such larger $h/J$. We therefore added a remark at the end of Section 2.1 where we now explicitly mention that we provide further data in the appendix and we discuss the time-window over which our approach provides a quantitatively accurate solution.

ii)

-- Referee: "In two-dimensions the transverse-field Ising model has been studied before in Refs.[20,21]. The authors should try to compare their numerical results with the previous one, if possible."

-- Response: While desirable, a quantitative comparison of the observable dynamics to the anticipated references is not possible, because our approach in its current formulation applies to a different parameter regime. In Refs. [20,21] quenches within the two phases are considered without crossing the underlying equilibrium quantum phase transition of the quantum Ising model. In our work we study the opposite case of crossing the transition. To emphasize these different parameter regimes, we added a note at the bottom of page 3. Furthermore, it is one of our main results, that the Jastrow classical network ansatz, which is used in Ref. [20], is missing an important Ising plaquette interaction in the classical network for the two-dimensional Ising model. From our perturbative approach we rather find that even to lowest order in perturbation theory further interaction terms have to be taken into account, as was already mentioned in the original version below Eq. (4).

iii)

-- Referee: "The results in the paper are about local observables, near-neighbour correlations and two-site entanglement. Can the method also be used to calculate long-range correlations, block-entanglement, etc.?"

-- Response: The sampling of long-range correlations does not constitute a principled problem with the classical network wavefunction. However, we find that long-range correlations are only obtained accurately if higher order expansions are employed, as mentioned on page 6 and in Appendix A.2. Performing high-order perturbative expansions become, of course, unfeasible at some point. Therefore, it appears more likely that for the purpose of capturing also long-range correlations accurately, it is more advantageous to augment our network structure with a time-dependent variational principle. In general, it is also possible to obtain the entanglement entropy for larger blocks than what is shown in our work. However, the approach of sampling all correlation functions to reconstruct the reduced density matrix becomes exponentially hard for increasing block sizes. But it remains an interesting question whether Monte-Carlo methods can be employed for this purpose instead, as we now also shortly discuss in Section 2.3.

---

## Round 4 · List of Changes

As requested by Referee 1:

1) We expanded significantly the explanation in the caption of Fig. 1, in order to make the content of the figure more easily accessible.
2) We expanded the explanation below Eq. (10), where we now discuss how it is possible to sample the partition function even in the presence of complex weights.
3) We added a sentence above Eq. (11) to clarify that observables sampled from the ANN will be identical with pCN results.
4) We added a legend to Fig. 6 and refrained from using different colors to avoid confusion.
5) We see from the Referee’s comment that we might have indicated in our manuscript that we have performed a rigorous complexity comparison. In order to avoid this impression, we removed the corresponding comment from the Conclusions section.

As requested by Referee 2:

1) We expanded the explanation in the caption of Fig. 1.
2) We added an explanation of “maximal bond dimension” in Appendix A.3.
3) We added a remark including these references below Eq. (2).

Due to comments by Referee 2:
1) We added a remark at the end of Section 2.1 mentioning the additional data for larger $h/J$ in the appendix and discussing the time-window over which our approach provides a quantitatively accurate solution.
2) We added a note at the bottom of page 3 commenting on the results for 2D TFIMs in Ref. [20,21].
3) We added a short discussion of how to compute entanglement of larger blocks in Section 2.3.

Unrelated changes:

1) We corrected a typo in Eq. (24)

You are currently on this page

Resubmission 1707.06656v4 on 24 January 2018

---

## Editorial Decision

published